# M-VAR: Decoupled Scale-wise Autoregressive Modeling for High-Quality Image Generation

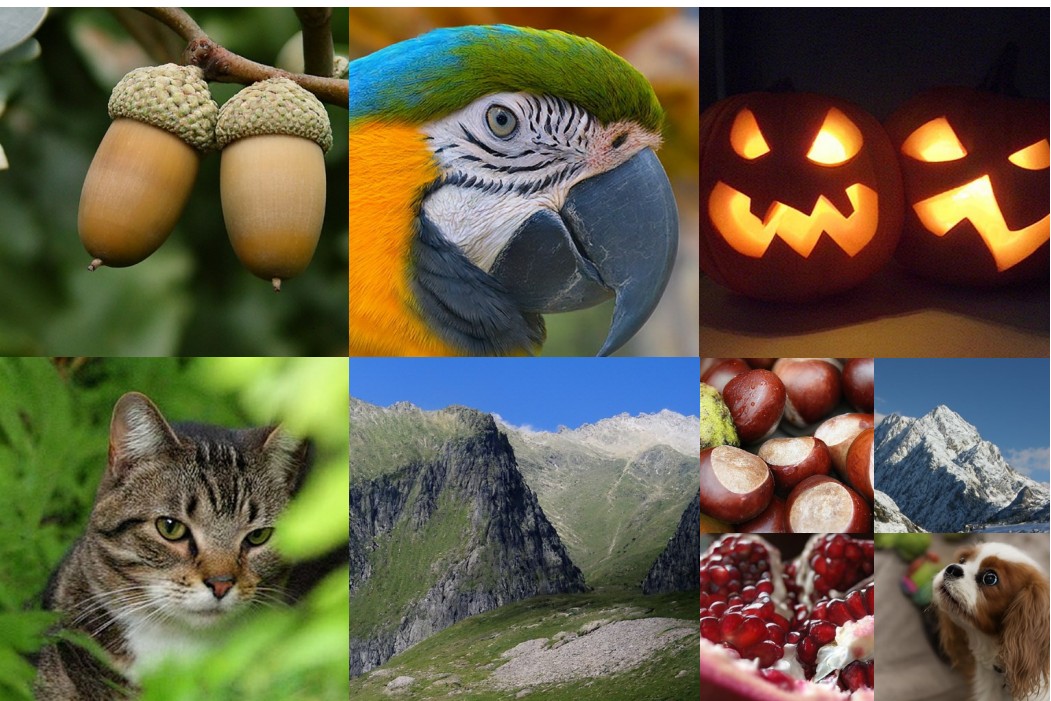

Figure 1: **Generated 512×512 and 256×256 samples from our M-VAR trained on ImageNet.**

## Abstract

There exists recent work in computer vision, named VAR, that proposes a new autoregressive paradigm for image generation. Diverging from the vanilla next-token prediction, VAR structurally reformulates the image generation into a coarse to fine next-scale prediction. In this paper, we show that this scale-wise autoregressive framework can be effectively decoupled into *intra-scale modeling*, which captures local spatial dependencies within each scale, and *inter-scale modeling*, which models cross-scale relationships progressively from coarse-to-fine scales. This decoupling structure allows to rebuild VAR in a more computationally efficient manner. Specifically, for intra-scale modeling — crucial for generating high-fidelity images — we retain the original bidirectional self-attention design to ensure comprehensive modeling; for inter-scale modeling, which semantically connects different scales but is computationally intensive, we apply linear-complexity mechanisms like Mamba to substantially reduce computational overhead. We term this new framework M-VAR. Extensive experiments demonstrate that our method outperforms existing models in both image quality and generation speed. For example, our 1.5B model, with fewer parameters and faster inference speed, outperforms the largest VAR-d30-2B. Moreover, our largest model M-VAR-d32 impressively registers 1.78 FID on ImageNet 256×256 and outperforms the prior-art autoregressive models LlamaGen/VAR by 0.4/0.19 and popular diffusion models LDM/DiT by 1.82/0.49, respectively.

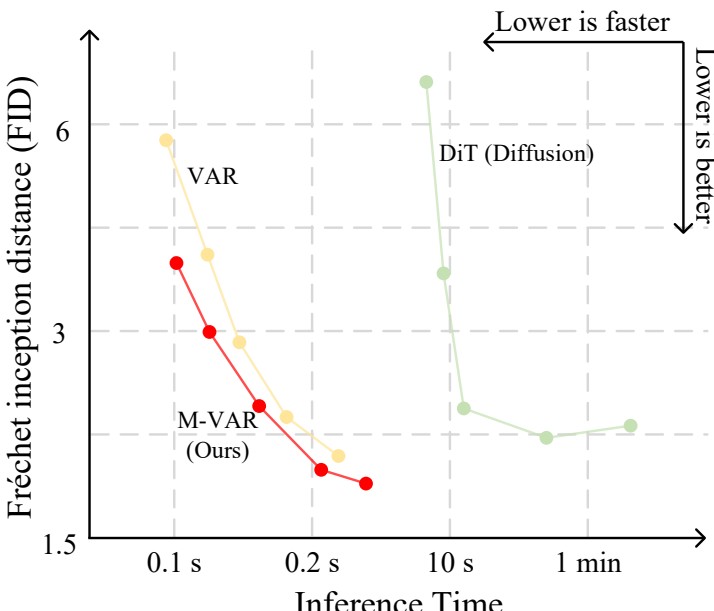

Figure 2: Fréchet inception distance (FID) on 256×256 image generation. Our M-VAR-1.5B model outperforms the largest 2B VAR-d30 with fewer parameters and faster inference speed. Our largest M-VAR-3B achieves 1.78 FID.

# 1 INTRODUCTION

Autoregressive models (Radford et al., 2018; Brown et al., 2020) have been instrumental in advancing the field of natural language processing (NLP). By modeling the probability distribution of a token given the preceding ones, these models can generate coherent and contextually relevant text. Prominent examples like GPT-3 (Brown et al., 2020) and its successors (OpenAI, 2022; 2023) have demonstrated remarkable capabilities in language understanding and generation, setting new benchmarks across various NLP applications.

Building upon the success in NLP, the autoregressive modeling paradigm (Yu et al., 2022; Sun et al., 2024; Van den Oord et al., 2016) has also been extended to computer vision for image generation tasks, aiming to generate high-fidelity images by predicting visual content in a sequential manner. Recently, VAR (Tian et al., 2024) has further enhanced this image autoregressive pipeline by structurally reformulating the learning target into a coarse-to-fine "next-scale prediction", which innately introduces strong semantics to interconnecting tokens along scales. As demonstrated in the VAR paper, this pipeline exhibits much stronger scalability and can achieve competitive, sometimes even superior, performance compared to advanced diffusion models.

This paper aims to further optimize VAR's computation structure. Our key insight lies in decoupling VAR's cross-scale autoregressive modeling into two distinct parts: intra-scale modeling and inter-scale modeling. Specifically, intra-scale modeling involves bidirectionally modeling multiple tokens within each scale, capturing intricate spatial dependencies and preserving the 2D structure of images. In contrast, inter-scale modeling focuses on unidirectional causality between scales by sequentially progressing from coarse to fine resolutions — each finer scale is generated conditioned on all preceding coarser scales, ensuring that global structures guide the refinement of local details. Notably, the sequence length involved in inter-scale modeling is much longer than that of intra-scale modeling, resulting in significantly higher computational costs. But meanwhile, our analysis of attention scores for both intra-scale and inter-scale interactions (as discussed in Sec. 3.2) suggests a contrasting reality: intra-scale interactions dominate the model's attention distribution, while inter-scale interactions contribute significantly less.

Motivated by the observations above, we propose to develop a more customized computation configuration for VAR. For the intra-scale component, given the much shorter sequence lengths within each

scale and its significant contribution to the model's attention distribution, we retain the bidirectional attention mechanism to fully capture comprehensive spatial dependencies. This ensures that local spatial relationships and fine-grained details are effectively modeled at a reasonable computational overhead. Conversely, for the inter-scale component, which involves much longer sequences but demands relatively less comprehensiveness in modeling global relationships, we adopt Mamba (Gu & Dao, 2023; Dao & Gu, 2024), a linear-complexity mechanism, to handle such inter-scale dependencies efficiently.

By segregating these two modeling modules and applying appropriate mechanisms to each, our approach significantly reduces computational complexity while preserving the model's ability to maintain 2D spatial coherence and unidirectional coarse-to-fine consistency, making it well-suited for high-quality image generation. As shown in Figure 2, our proposed framework, which we term M-VAR, outperforms existing models in both image quality and inference speed. For instance, our 1.5B parameter M-VAR model achieves an FID score of 1.93 with fewer parameters and 1.2× faster inference speed, outperforming the largest VAR model, which uses 2B parameters and attains an FID score of 1.97. Moreover, our largest model, M-VAR-d32, achieves an impressive FID score of 1.78 on ImageNet at $256 \times 256$ resolution, outperforming the previous best autoregressive models LlamaGen by 0.4 and VAR by 0.19, respectively, and well-known diffusion models LDM by 1.82 and DiT by 0.49, respectively.

## 2 RELATED WORK

### 2.1 VISUAL GENERATION

Visual Generation can generally be split into three categories: 1) Diffusion models (Dhariwal & Nichol, 2021; Rombach et al., 2022) treat visual generation as the reverse process of the diffusion process. 2) Mask prediction model (Chang et al., 2022) follows BERT-style (Devlin, 2018) language model to generate images by predicting mask tokens 3) Autoregressive models generate images by predicting the next pixel/token/scale in a sequence. We focus on the last one in this paper.

The pioneering method that brings autoregressive into visual generation is PixelCNN (Van den Oord et al., 2016), which models images by predicting the discrete probability distributions of raw pixel values, effectively capturing all dependencies within an image. Building on this foundation, VQGAN (Esser et al., 2021) advances the field by applying autoregressive learning within the latent space of VQVAE (Razavi et al., 2019), simplifying the data representation for more efficient modeling. The RQ Transformer (Lee et al., 2022) introduces a novel technique using a fixed-size codebook to approximate an image's feature map with stacked discrete codes, forecasting the next quantized feature vectors by predicting subsequent code stacks. Parti (Yu et al., 2022) takes a different route by framing image generation as a sequence-to-sequence modeling task akin to machine translation, using sequences of image tokens as targets instead of text tokens, and thus capitalizing on the significant advancements made in large language models through data and model scaling. LlamaGen (Sun et al., 2024) further extends this concept by applying the traditional "next-token prediction" paradigm of large language models to visual generation, demonstrating that standard autoregressive models like Llama can achieve state-of-the-art image generation performance when appropriately scaled, even without specific inductive biases for visual signals. Lastly, VAR (Tian et al., 2024) reimagines autoregressive learning for images by adopting a coarse-to-fine strategy termed "next-scale prediction" departing from the conventional raster-scan "next-token prediction" method to offer a new perspective on image generation.

### 2.2 MAMBA

State-space models (SSMs) (Gu et al., 2021a;b) have recently emerged as a compelling alternative to Convolutional Neural Networks (CNNs) (LeCun et al., 1998) and Transformers (Vaswani, 2017) for capturing long-range dependencies with linear computational complexity. These models employ hidden states to represent sequences efficiently. The latest advancement in this domain is Mamba (Gu & Dao, 2023; Dao & Gu, 2024), a sophisticated SSM that introduces data-dependent layers with expanded hidden states. Mamba constructs a versatile language model backbone that not only rivals Transformers across various scales but also maintains linear scalability with respect to sequence length.

Building on Mamba's success in natural language processing, its application has been extended to computer vision tasks. Vision Mamba (Vim) (Zhu et al., 2024) utilizes pure Mamba layers within Vim blocks, leveraging both forward and backward scans to model bidirectional representations. This approach effectively addresses the direction-sensitive limitations inherent in the original Mamba model. Additionally, ARM (Ren et al., 2024) pioneers the integration of autoregressive pretraining with Mamba in the vision domain.

In the realm of image generation, Diffusion Mamba (DiM) (Fei et al., 2024) combines the efficiency of the Mamba sequence model with diffusion processes to achieve high-resolution image synthesis. DiM employs multi-directional scans, introduces learnable padding tokens, and enhances local features to adeptly manage two-dimensional signal processing. AiM (Li et al., 2024) further advances this by replacing Transformers with Mamba for autoregressive image generation, following methodologies similar to LlamaGen (Sun et al., 2024). However, these existing methods typically apply Mamba to sequences with lengths up to 256. Our proposed M-VAR model extends this capability by using Mamba to capture inter-scale dependencies in sequences as long as 2,240 tokens. This significant increase in sequence length underscores Mamba's efficiency and effectiveness in modeling long sequences within vision applications.

## 3 METHOD

### 3.1 PRELIMINARY: AUTOREGRESSIVE MODELING

**Autoregressive modeling in natural language processing.** Given a set of corpus $\mathcal{U} = \{u_1, ..., u_n\}$, autoregressive modeling predicts next words based on all preceding words:

$$p(u) = \prod_{i=1}^{n} p(u_i | u_1, ..., u_{i-1}, \Theta) \tag{1}$$

Autoregressive modeling minimize the negative log-likelihood of each word $u_i$ given all preceding words from $u_1$ to $u_{i-1}$:

$$\mathcal{L} = -\log p(u) \tag{2}$$

This strategy leads to the success of a large language model.

**Token-wise autoregressive modeling in computer vision.** From language to image, to apply autoregressive pertaining, image tokenization via vector-quantization transfers 2D images $X \in \mathcal{R}^{H \times W \times C}$ 2D tokens and flatten tokens into 1D token sequences $X = \{x_1, x_2, ..., x_n\}$:

$$\mathcal{L} = -\sum_{i=1}^{N} \log p(x_i | x_1, ..., x_{i-1}, \Theta) \tag{3}$$

However, such the flatten operation breaks down the 2D structure of an image. Therefore, VAR (Tian et al., 2024) proposes to perform scale-wise autoregressive modeling to keep the 2D structure.

**Scale-wise autoregressive modeling.** Instead of tokenizing image into a sequence of tokens, VAR tokenizes the image into multi-scale token maps $S = \{s_1, ..., s_n\}$, where $s_i$ is the token map with the resolution of $h_i \times w_i$ downsampling from $s_n \in \mathcal{R}^{h_n \times w_n}$, therefore, $s_i$ contains $h_i \times w_i$ tokens and maintains the 2D structure, while $x_i$ only contain one token and break the 2D structure. The auto regressive model is reformed to:

$$\mathcal{L} = -\sum_{i=1}^{N} \log p(s_i | s_1, ..., s_{i-1}, \Theta) \tag{4}$$

In practice, the sequence $S$ of multiple scales is much longer than each scale $(s_1, ..., s_n)$. VAR utilizes attention and Transformer to implement this algorithm. For generating the $i_{th}$ scale, VAR attends the first scale to the $(i-1)_{th}$ and generates $h_i \times w_i$ tokens in parallel as the $i_{th}$ scale rather than token by token.

| Attention Mode | Attention Score | Computation Cost |
|---|---|---|
| *256×256 Image Generation* | | |
| Intra Scale | 79.6% | 23.9% |
| Inter Scale | 20.4% | 76.1% |
| *512×512 Image Generation* | | |
| Intra Scale | 77.1% | 30.3% |
| Inter Scale | 22.9% | 69.7% |

Table 1: The statistics of attention score and computation cost of the attention in VAR.

## 3.2 DECOUPLE SCALE-WISE AUTOREGRESSIVE MODELING

We can break the attention in VAR into two parts: 1) bidirectionally attend intra scale which the sequence length is much shorter; 2) Uni-directional attend from coarse-scale to fine scale which the sequence length is much longer.

We show the statistic of attention score and the computation cost of VAR in Table 1. Surprisingly, intra-scale attention scores account for 79.6% of the total attention scores in 256×256 image generation and 77.1% in 512×512 image generation. This dominance of intra-scale attention suggests that capturing fine-grained details within the same scale is crucial for high-quality image synthesis. However, a closer examination of the computation cost presents a contrasting scenario. Despite intra-scale attention contributing the most to the attention scores, it only consumes 23.9% of the computation cost for 256×256 images and 30.3% for 512×512 images. In stark contrast, inter-scale attention, which accounts for a smaller portion of the attention scores (20.4% and 22.9% for 256×256 and 512×512 images respectively), is responsible for the majority of the computation cost—76.1% and 69.7% respectively. The disparity between the attention scores and computation cost highlights an inefficiency in the current attention mechanism in VAR.

Based on this observation, we propose a novel approach to optimize the efficiency of the scale-wise autoregressive image generation model. Specifically, we use standard attention mechanisms for intra-scale interactions—where the majority of attention is naturally focused, and computation is relatively low and employ Mamba, a model with linear computational complexity, for inter-scale interactions. By integrating Mamba for inter-scale attention, we aim to significantly reduce the computational overhead without compromising the model's ability to capture essential cross-scale dependencies. Mamba is designed to handle long-range interactions efficiently that scales linearly with the sequence length, as opposed to the quadratic scaling of traditional attention mechanisms. This makes it particularly suitable for modeling inter-scale relationships, where the computational cost is otherwise prohibitive.

As shown in Figure 3, our proposed M-VAR introduces an efficient approach for scale-wise autoregressive image generation by combining traditional attention mechanisms with Mamba, a model characterized by linear computational complexity. Given an image with multiple scales $S = [s_1, ..., s_n]$, we aim to model both intra-scale and inter-scale representations effectively while optimizing computational.

To capture the fine-grained details and local dependencies within each scale, we apply an attention mechanism independently to each scale:

$$S' = [s'_1, ..., s'_n] = [Attn(C), Attn(Upsample(s_1)), ..., Attn(Upsample(s_{n-1}))] \quad (5)$$

Here, $Attn$ represents the attention applied to scale, producing the intra-scale representation and $C$ is the condition token. All attention share the same parameters but process each scale independently. This design choice ensures consistency across scales and reduces the overall model complexity. For efficient implementation, we adopt FlashAttention (Dao et al., 2022; Dao, 2024) to perform the intra-scale attention in parallel.

After obtaining the intra-scale representations $S'$, modeling the relation between different scales becomes crucial for ensuring global coherence and coarse-to-fine consistency in the generated images. However, as previously discussed, traditional attention mechanisms are computationally expensive for inter-scale interactions due to their quadratic complexity, and we adopt the Mamba model with

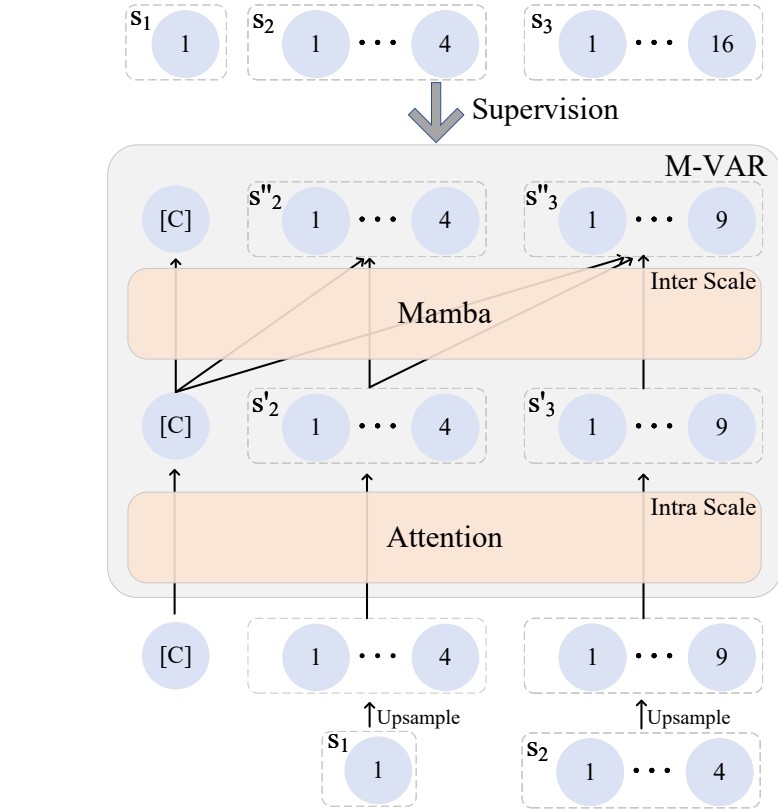

Figure 3: An overview of M-VAR. M-VAR takes the input sequence of $\{[C], s_1, ..., s_{n-1}\}$ to predict $\{s_1, ..., s_n\}$ where $[C]$ is the condition token. The model first divides the input into different scales and applies an attention mechanism to capture intra-scale spatial correlations. It then utilizes Mamba to autoregressively model inter-scale dependencies, enabling coherent and efficient multi-scale image generation.

linear complexity.

$$S'' = [s_1'', ..., s_n''] = Mamba(Concat([s_1', ..., s_n']))\tag{6}$$

By concatenating $s'$ from all scales into a single sequence, Mamba efficiently processes the combined representations, capturing the essential inter-scale interactions without the heavy computational burden.

## 4 EXPERIMENT

### 4.1 IMAGENET 256×256 CONDITIONAL GENERATION

Following the same settings (Tian et al., 2024), we train M-VAR on ImageNet (Deng et al., 2009) for 256×256 conditional generation. We design multiple model variants with depths of 12, 16, 20, 24, and 32 layers.

We compare our M-VAR with previous state-of-the-art generative adversarial nets(GAN), diffusion models, autoregressive models, mask prediction models. As shown in Table 2, Our M-VAR models offer a balanced synergy of high image quality and computational efficiency. M-VAR outperforms GANs in terms of image fidelity and diversity while maintaining comparable inference speeds. Compared to diffusion models, M-VAR models deliver superior or comparable image quality with significantly reduced inference time. Against token-wise autoregressive and mask prediction models, our models achieve better performance metrics with fewer steps and faster inference times.

Table 2: **Generative model comparison on class-conditional ImageNet 256×256**. Metrics include Fréchet inception distance (FID), inception score (IS), precision (Pre) and recall (rec). Step: the number of model runs needed to generate an image. Time: the relative inference time of M-VAR.

| Model | FID↓ | IS↑ | Pre↑ | Rec↑ | Param | Step | Time |
|---|---|---|---|---|---|---|---|
| *Generative Adversarial Net (GAN)* | | | | | | | |
| BigGAN (Brock et al., 1809) | 6.95 | 224.5 | 0.89 | 0.38 | 112M | 1 | – |
| GigaGAN (Kang et al., 2023) | 3.45 | 225.5 | 0.84 | 0.61 | 569M | 1 | – |
| StyleGan-XL (Sauer et al., 2022) | 2.30 | 265.1 | 0.78 | 0.53 | 166M | 1 | 0.2 |
| *Diffusion* | | | | | | | |
| ADM (Dhariwal & Nichol, 2021) | 10.94 | 101.0 | 0.69 | 0.63 | 554M | 250 | 118 |
| CDM (Ho et al., 2022) | 4.88 | 158.7 | – | – | – | 8100 | – |
| LDM-4-G (Rombach et al., 2022) | 3.60 | 247.7 | – | – | 400M | 250 | – |
| DiT-L/2 (Peebles & Xie, 2023) | 5.02 | 167.2 | 0.75 | 0.57 | 458M | 250 | 2 |
| DiT-XL/2 (Peebles & Xie, 2023) | 2.27 | 278.2 | 0.83 | 0.57 | 675M | 250 | 2 |
| L-DiT-3B (dit, 2024) | 2.10 | 304.4 | 0.82 | 0.60 | 3.0B | 250 | >32 |
| L-DiT-7B (dit, 2024) | 2.28 | 316.2 | 0.83 | 0.58 | 7.0B | 250 | >32 |
| *Mask Prediction* | | | | | | | |
| MaskGIT (Chang et al., 2022) | 6.18 | 182.1 | 0.80 | 0.51 | 227M | 8 | 0.4 |
| RCG (cond.) (Li et al., 2023) | 3.49 | 215.5 | – | – | 502M | 20 | 1.4 |
| *Token-wise Autoregressive* | | | | | | | |
| VQVAE-2[†] (Razavi et al., 2019) | 31.11 | ∼45 | 0.36 | 0.57 | 13.5B | 5120 | – |
| VQGAN[†] (Esser et al., 2021) | 18.65 | 80.4 | 0.78 | 0.26 | 227M | 256 | 7 |
| VQGAN (Esser et al., 2021) | 15.78 | 74.3 | – | – | 1.4B | 256 | 17 |
| ViTVQ (Yu et al., 2021) | 4.17 | 175.1 | – | – | 1.7B | 1024 | >17 |
| RQTran. (Lee et al., 2022) | 7.55 | 134.0 | – | – | 3.8B | 68 | 15 |
| LlamaGen-3B (Sun et al., 2024) | 2.18 | 263.33 | 0.81 | 0.58 | 3.1B | 576 | - |
| *Scale-wise Autoregressive* | | | | | | | |
| VAR-$d12$ (Tian et al., 2024) | 5.81 | 201.3 | 0.81 | 0.45 | 132M | 10 | 0.2 |
| M-VAR-$d12$ | 4.19 | 234.8 | 0.83 | 0.48 | 198M | 10 | 0.2 |
| VAR-$d16$ (Tian et al., 2024) | 3.55 | 280.4 | 0.84 | 0.51 | 310M | 10 | 0.2 |
| M-VAR-$d16$ | 3.07 | 294.6 | 0.84 | 0.53 | 464M | 10 | 0.2 |
| VAR-$d20$ (Tian et al., 2024) | 2.95 | 302.6 | 0.83 | 0.56 | 600M | 10 | 0.3 |
| M-VAR-$d20$ | 2.41 | 308.4 | 0.85 | 0.58 | 900M | 10 | 0.4 |
| VAR-$d24$ (Tian et al., 2024) | 2.33 | 312.9 | 0.82 | 0.59 | 1.0B | 10 | 0.5 |
| M-VAR-$d24$ | 1.93 | 320.7 | 0.83 | 0.59 | 1.5B | 10 | 0.6 |
| VAR-$d30$ (Tian et al., 2024) | 1.97 | 323.1 | 0.82 | 0.59 | 2.0B | 10 | 0.7 |
| M-VAR-$d32$ | 1.78 | 331.2 | 0.83 | 0.61 | 3.0B | 10 | 1 |

Compared with the most related VAR, our proposed M-VAR models demonstrate significant advancements in both performance and efficiency. Across various depths, M-VAR consistently achieves lower Fréchet Inception Distance (FID) scores and higher Inception Scores (IS), indicating superior image quality and diversity. Specifically, M-VAR-d24 attains an FID of 1.93 and an IS of 320.7 with 1.5 billion parameters. M-VAR-d24 surpasses the largest VAR model, VAR-d30, with 25% fewer parameters and 14% faster inference speed. Furthermore, our largest model, M-VAR-d32, achieves state-of-the-art performance with an FID of 1.78 and an IS of 331.2, utilizing 3.0 billion parameters. These results highlight the effectiveness of our approach in integrating intra-scale attention with Mamba for inter-scale modeling, leading to superior image generation quality and computational efficiency compared to existing models. The consistent outperformance of M-VAR models underscores their potential for scalable, high-resolution image generation. We also show more qualitative results in Figure 4.

As shown in Table 3, we also compare our M-VAR-d32 model with other state-of-the-art methods using rejection sampling on class-conditional ImageNet 256×256. Our M-VAR-d32 achieves an FID of 1.63 and an IS of 361.5, outperforming all compared models. Specifically, it surpasses the previous best VAR-d30 by FID of 0.1 and IS of 11.3. Additionally, M-VAR-d32 demonstrates significant improvements over ViTVQ, RQTransformer, and VQGAN by FID of 1.41, 2.17, 3.57

Table 3: **Generative model comparison on class-conditional ImageNet 256×256 with rejection sampling**.

| Model | Params | FID↓ | IS↑ |
|---|---|---|---|
| ViTVQ (Yu et al., 2021) | 1.7B | 3.04 | 227.4 |
| RQTran. (Lee et al., 2022) | 3.8B | 3.80 | 323.7 |
| VQGAN (Esser et al., 2021) | 1.4B | 5.20 | 280.3 |
| VAR-d30 (Tian et al., 2024) | 2.0B | 1.80 | 343.2 |
| M-VAR-d32 | 3.0B | 1.67 | 361.5 |

Table 4: **Generative model comparison on class-conditional ImageNet 512×512**.

| Model | FID↓ | IS↑ | Inference Time↓ |
|---|---|---|---|
| BigGAN (Brock et al., 1809) | 6.95 | 224.5 | 1 |
| DiT-XL/2 (Peebles & Xie, 2023) | 3.04 | 240.8 | 160 |
| MaskGiT (Chang et al., 2022) | 7.32 | 156.0 | 1 |
| VQGAN (Esser et al., 2021) | 26.52 | 66.8 | 50 |
| VAR-d36 (Tian et al., 2024) | 2.63 | 303.2 | 2 |
| M-VAR-d24 | 2.65 | 305.1 | 1 |

respectively. These results highlight the effectiveness of our approach in achieving superior image generation quality under rejection sampling, affirming the advancements of M-VAR in the realm of scale-wise autoregressive image generation.

## 4.2 IMAGENET 512×512 CONDITIONAL GENERATION

We train M-VAR on ImageNet (Deng et al., 2009) for 512×512 conditional generation. As shown in Table 4, our M-VAR-d24 model exhibits competitive performance in class-conditional ImageNet 512×512 generation when compared to the state-of-the-art generative approaches, VAR. Specifically, M-VAR-d24 achieves an FID of 2.65 and an IS of 305.1, closely matching the performance of VAR-d36. Importantly, M-VAR-d24 accomplishes this with half the inference time of VAR-d36, highlighting the efficiency gains from our decoupled intra-scale and inter-scale modeling approach. Compared to other generative models, such as BigGAN, DiT-XL/2, MaskGIT, and VQGAN, M-VAR-d24 consistently outperforms them in both FID and IS metrics while maintaining a lower or comparable inference time. We also show more qualitative results in Figure 4. The figures highlight that M-VAR consistently produces images with fine details, enhanced texture fidelity, and great structural coherence.

## 4.3 ABLATION STUDY

**Parameters.** We reduce M-VAR's parameters by adjusting its width or depth, aiming for a fair comparison while assessing the impact on performance and computational efficiency. As shown in Table 5, we present three variants of M-VAR alongside the baseline VAR model under similar parameter constraints. Firstly, M-VAR-W reduces the width of the model from 1024 to 768 while keeping the depth constant at 16 layers. This reduction leads to a decrease in the total number of parameters to 260 million, which is lower than VAR's 310 million parameters. Remarkably, even with fewer parameters, M-VAR-W achieves a better FID score of 3.20 compared to VAR's 3.55, indicating an improvement in image generation quality. Additionally, the training cost is reduced to 0.9 times that of VAR, showcasing enhanced efficiency. Similarly, M-VAR-D maintains the original width of 1024 but reduces the depth from 16 to 12 layers. M-VAR-D attains an FID score of 3.19, outperforming VAR while also reducing the training cost and inference time to 0.8 times that of the VAR. These results illustrate that our proposed M-VAR models can achieve superior image generation quality compared to the baseline VAR, even when operating under similar or reduced parameter budgets.

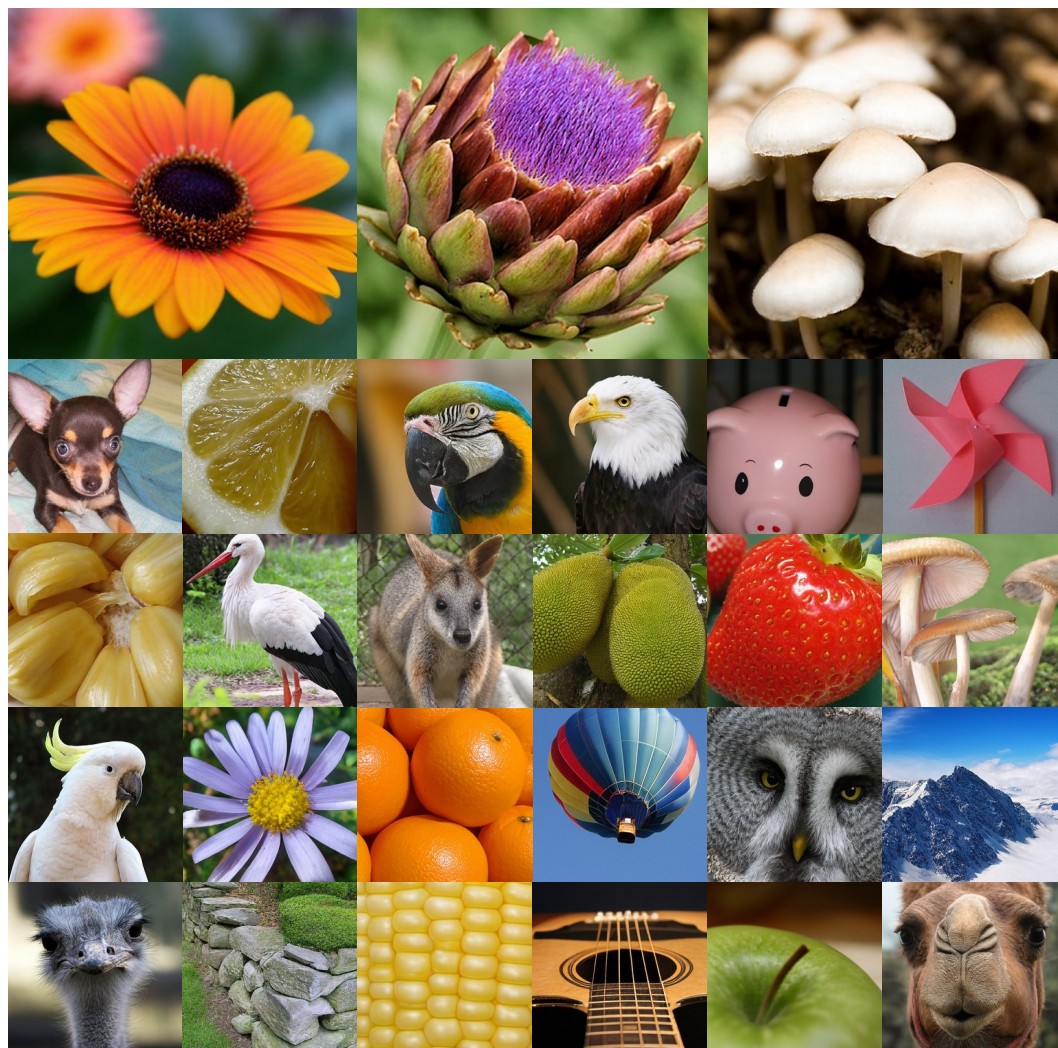

Figure 4: **Qualitative Results.** We show the images generated by our M-VAR.

Table 5: Compare with VAR under similar parameters. † our default settings.

| Model | Depth | With | Param. | FID↓ | Training Cost ↓ | Inference Time↓ |
|---|---|---|---|---|---|---|
| VAR | 16 | 1024 | 310M | 3.55 | 1 | 0.9 |
| M-VAR-W | 16 | 768 | 260M | 3.20 | 0.9 | 0.9 |
| M-VAR-D | 12 | 1024 | 340M | 3.19 | 0.8 | 0.7 |
| M-VAR† | 16 | 1024 | 450M | 3.07 | 1 | 1 |

**From VAR to MAR.**    We gradually replaced the global attention layers in VAR with our proposed intra-scale attention and inter-scale Mamba modules to evaluate their impact on image generation quality. As shown in Table 5, we incrementally increased the number of layers replaced—from 0 in the original VAR model to all 16 layers in our model. The results demonstrate a consistent improvement in FID scores from 3.55 to 3.07 as more global attention layers are replaced. The improvements suggest that decoupling the modeling of intra-scale and inter-scale dependencies positively impacts image synthesis quality. By effectively capturing local spatial details within each scale and efficiently modeling hierarchical relationships between scales, our approach leads to more coherent and detailed image generation.

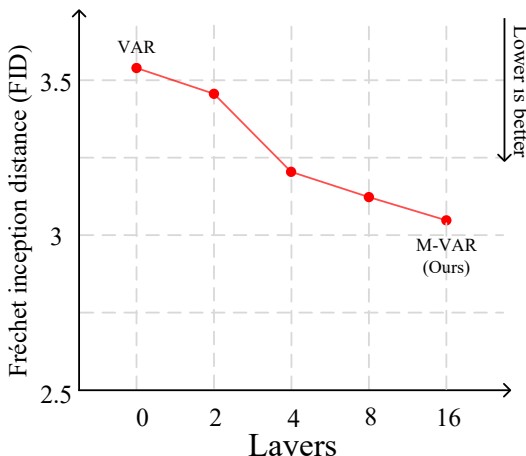

Figure 5: The effectiveness of our decouple design. We gradually replace the global attention with our intra-scale attention and inter-scale mamba.

Table 6: Effectiveness and efficiency of Attention and Mamba. We compare our intra-scale attention and Mamba with previous global attention in VAR

| Method | Global Attention | Intra-scale Attention | Mamba | FID ↓ |
|---|---|---|---|---|
| VAR | ✓ | | | 3.55 |
| 1 | | ✓ | | 7.17 |
| 2 | | | ✓ | 4.12 |
| M-VAR (Ours) | | ✓ | ✓ | 3.07 |

**Effectiveness and efficiency of Attention and Mamba.** Table 6 illustrates the impact of different attention mechanisms on image generation quality, as measured by the Fréchet Inception Distance (FID). The baseline VAR model employs global attention, capturing both intra-scale and inter-scale dependencies simultaneously, and achieves an FID of 3.55. When using only intra-scale attention without inter-scale modeling (Method 1), the FID significantly deteriorates to 7.17, indicating that inter-scale dependencies are crucial for high-quality image generation. Method 2, which also utilizes intra-scale attention but includes some enhancements, improves the FID to 4.12, yet still falls short of the baseline VAR performance. Our proposed M-VAR model combines intra-scale attention with Mamba for efficient inter-scale modeling. By decoupling the two types of dependencies and applying Mamba's linear-complexity approach for inter-scale interactions, M-VAR achieves the best FID of 3.07. This demonstrates that effectively capturing intra-scale dependencies with attention and efficiently modeling inter-scale relationships with Mamba leads to superior image quality.

## 5 CONCLUSION

We propose a novel approach to scale-wise autoregressive image generation that decouples intra-scale and inter-scale modeling to enhance both efficiency and performance. By employing bidirectional attention mechanisms for intra-scale interactions, our model effectively captures detailed spatial dependencies within each scale without excessive computational overhead. For inter-scale modeling, we utilize Mamba with linear complexity, addressing the disproportionate computational burden typically associated with inter-scale attention mechanisms. This strategic separation allows our model to maintain spatial coherence and hierarchical consistency while significantly reducing computational complexity. Our experiments demonstrate that this decoupled framework outperforms existing autoregressive and diffusion models, achieving superior image quality with fewer parameters and faster inference speeds.

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
