# OpenReview forum: "M-VAR: Decoupled Scale-wise Autoregressive Modeling for High-Quality Image Generation"
_ICLR.cc/2025/Conference — ICLR 2025 Conference Withdrawn Submission_

### Official Review · Reviewer_qR12 · 2024-11-03

**Soundness:** 2
**Presentation:** 3
**Contribution:** 2
**Rating:** 3
**Confidence:** 5

**Summary:**

This paper introduces M-VAR, a new autoregressive image model based on VAR. The core idea is to decouple VAR into intra-scale modeling and inter-scale modeling. For intra-scale modeling, softmax attention is used, while mamba is used for inter-scale modeling. On ImageNet 256x256 and ImageNet 512x512, M-VAR achieves better efficiency/FID than VAR.

**Strengths:**

1. This paper is well-written. The motivation is clear and reasonable. The proposed method is also presented clearly.
2. On ImageNet 256x256 and 512x512, M-VAR demonstrates better results than VAR.
3. Ablation study shows intra-scale attention + mamba works better than global attention on VAR.

**Weaknesses:**

1. Technical contribution is limited. Replacing global attention with hybrid model architectures has been extensively explored in the AI community. A big concern of such designs is that they may not preserve advantages after scaling up and applying them to real-world use cases (e.g., text-to-image generation). Given that this work only has ImageNet results, the value of the current manuscript is limited for the community.
2. It is unclear why M-VAR can deliver better FID than VAR. From the model capacity perspective, global attention should have a stronger/similar capacity than intra-scale attention and mamba.
3. Current design choices seem quite random, lacking detailed ablation studies. For example, there are many different choices for intra-scale modeling and inter-scale modeling (RWKV, linear attention, etc). Is there any insight on why choosing the current design?
4. According to the ImageNet experiments, the improvements look a bit incremental.

**Questions:**

1. What's the setting for speed comparison (hardware, inference engine, batch size, etc)? In addition to relative speedup ratios, adding measured latency/throughput in the tables will be better.
2. Why M-VAR can deliver better FID than VAR? I can see that M-VAR has advantages over VAR from the efficiency perspective. But, from the model capacity perspective, I do not see clear advantages.

---

### Official Review · Reviewer_aiUi · 2024-11-04

**Soundness:** 2
**Presentation:** 3
**Contribution:** 2
**Rating:** 5
**Confidence:** 3

**Summary:**

This paper proposes a hybrid framework combining Mamba and Attention mechanisms for scale-wise autoregressive image generation. While the approach appears standard, the claim made by the authors is that the decoupling of intra-scale and inter-scale modeling improves computational efficiency and image quality.

**Strengths:**

1. The paper is generally well-written and easy to follow.
2. It includes numerous objective metrics that contribute to the evaluation.
3. Observing Table 1, it is evident that reducing intra-scale attention operations is necessary due to the computational cost highlighted.

**Weaknesses:**

1. The presentation could be improved as some figures, such as Figures 2, 3, and 5, are overly large and impact readability.
2. Despite the inclusion of many metrics, several tables exhibit issues:
    * In Table 2, under the section "Generative model comparison," the comparison between Scale-wise Autoregressive models (M-VAR and VAR) seems unfair. For example, the last two rows show that M-VAR (depth 32) with 3B parameters outperforms VAR (depth 30) with 2B parameters, but the parameter count for M-VAR is 50% higher.
    * Additionally, inference time increases from 0.7s to 1s (a 43% increase) despite only slightly better FID and IS scores.
3. Table 6 appears to lack significant information and could be made more concise for clarity.
4. It is suggested that the data in Table 1 be illustrated as a figure to better highlight this critical motivation behind the work.

**Questions:**

Was the VAR-d36 model trained by the authors since it has not been released? (in Table 4)

---

### Official Review · Reviewer_duhF · 2024-11-04

**Soundness:** 2
**Presentation:** 1
**Contribution:** 3
**Rating:** 3
**Confidence:** 5

**Summary:**

This work builds upon the prior work VAR [1] model for autoregressive multiscale image generation. The work shows that inter-scale dependencies have higher computational cost compared to intra-scale dependencies and extends the inter-scale attention mechanism with Mamba-like attention. Experiments on ImageNet 256 and class-conditional 512 show that model performs better than VAR in terms on the FID and IS scores.

[1] Visual Autoregressive Modeling: Scalable Image Generation via Next-Scale Prediction

**Strengths:**

+ The proposed approach provides statistics on the computational overhead of the intra-scale and inter-scale attention modules for autoregressive multiscale image generation. The statistics are used to design a new mamba based attention module of modeling inter-scale dependencies
+ Adequate experiments and ablations are performed that show better FiD and IS compared to prior work.

**Weaknesses:**

- The paper is very difficult to read.  In eq. 2, the parametrization \theta is not defined. ll. 201-202 are not correct. There are a lot of broken sentences and grammatical ill-constructed sentences. For example, ll. 213 "The sequence S of multiple scales is much longer than each scale (s1, ..., sn)" is not clear. ll. 229 -231 are broken.

- What is meant by the attention score, reported in Table 1. how is this score computed is not defined or explained.

- Images have a local dependency structure.  Therefore intra-scale dependencies are easier to model. It will be good to provide an evidence with the pixel correlations on the dataset considered as a function of inter-pixel distance.

- In table 4, how is the inference time of M-VAR lower compared to the VAR model while in table 5 its slightly higher or comparable. The paper mentions that the reduction is quadratic in computational efficiency. How do these results demonstrate the effect?

- The number of parameters for the proposed model are much higher than the baseline VAR model. The work claims to improve the computational cost of the baseline. How do these results justify the claim.

- Prior work [a,b,c,d], also performs multi-scale image generation. How does this approach compare to the prior work? A line work exists on multi-scale image generation with autoregressive models. The related work does not discuss the prior work for multi-scale image generation.

[a] Mahajan, Shweta and Roth, Stefan. PixelPyramids: Exact Inference Models from Lossless Image Pyramids. In ICCV,  2021.

[b] Xuezhe Ma, Xiang Kong, Shanghang Zhang, and Eduard H.Hovy. MaCow: Masked convolutional generative flow. In NeurIPS, 2019.

[c] Jacob Menick and Nal Kalchbrenner. Generating high fidelity images with subscale pixel networks and multidimensional upscaling. In ICLR, 2019.

[d] Scott E. Reed, Aäron van den Oord, Nal Kalchbrenner, Sergio Gomez Colmenarejo, Ziyu Wang, Yutian Chen, Dan
Belov, and Nando de Freitas. Parallel multiscale autoregressive density estimation. In ICML,  2017.

**Questions:**

- What is attention score and how is it computed? Are these results on the test images? If so, how many images are considered for the statistics?
- How does the increased number of parameters correlated with the claimed computational efficiency of the model.
- Also see weaknesses above, for additional reviewer questions and concerns.

---

### Official Review · Reviewer_sAvT · 2024-11-04

**Soundness:** 3
**Presentation:** 3
**Contribution:** 3
**Rating:** 5
**Confidence:** 3

**Summary:**

This paper presents a new autoregressive image generation framework, M-VAR, which leverages bidirectional self-attention for intra-scale modeling and the Mamba mechanism for inter-scale modeling.
The proposed method seems improved both the computational efficiency and the quality of generated images compared to VAR.

**Strengths:**

(1) The statistics of attention score and computation cost of the attention in VAR are interesting and inspiring.

(2) The combination of intra-scale self-attention and inter-scale linear modeling seems a reasonable solution to improve the computational efficiency of VAR.

(3) The largest model of M-VAR achieves SOTA FIDs on ImageNet dataset.

**Weaknesses:**

(1) The decoupling of scale-wise autoregressive modeling seems reasonable, but why we must adopt Mamba? Other efficient self-attention variants should also be considered.

(2) In Table 2, M-VAR-dX usually has more parameters than VAR-dX. Are these additional parameters help M-VAR for better performance?

(3) The computational FLOPS are not discussed in this article, since the number of paramters is not the only factor affecting computational efficiency.

(4) The curve shown in Figure 5 might seem counter-intuitive, why global attention performances worse despite its global modeling capacity?

(5) Some typos, e.g. L480 'As shown in Table 5', it should be 'Figure 5'?

**Questions:**

see weaknesses

---

### Note · Authors · 2024-11-12

I have read and agree with the venue's withdrawal policy on behalf of myself and my co-authors.